# Tire Slip *H*_∞_ Control for Optimal Braking Depending on Road Condition

**DOI:** 10.3390/s23031417

**Published:** 2023-01-27

**Authors:** Miguel Meléndez-Useros, Manuel Jiménez-Salas, Fernando Viadero-Monasterio, Beatriz López Boada

**Affiliations:** Mechanical Engineering Department, Universidad Carlos III de Madrid, Avda. de la Universidad 30, 28911 Leganés, Spain

**Keywords:** tire slip control, vehicle dynamics, *H*_∞_ control, anti-lock brake system

## Abstract

Tire slip control is one of the most critical topics in vehicle dynamics control, being the basis of systems such the Anti-lock Braking System (ABS), Traction Control System (TCS) or Electronic Stability Program (ESP). The highly nonlinear behavior of tire–road contact makes it challenging to design robust controllers able to find a dynamic stable solution in different working conditions. Furthermore, road conditions greatly affect the braking performance of vehicles, being lower on slippery roads than on roads with a high tire friction coefficient. For this reason, by knowing the value of this coefficient, it is possible to change the slip ratio tracking reference of the tires in order to obtain the optimal braking performance. In this paper, an $H_{\infty }$ controller is proposed to deal with the tire slip control problem and maximize the braking forces depending on the road condition. Simulations are carried out in the vehicular dynamics simulator software CarSim. The proposed controller is able to make the tire slip follow a given reference based on the friction coefficient for the different tested road conditions, resulting in a small reference error and good transient response.

## 1. Introduction

Vehicle stability under braking is essential to ensure the integrity of the vehicle’s passengers and external actors. Wheel locking can affect vehicular motion, diverting the vehicle from the driver’s desired trajectory or reducing the effectiveness of braking, which can lead to accidents. In many cases, these accidents and their consequences can be avoided thanks to the use of active vehicle dynamics control systems.

Tire slip control by means of Anti-lock Braking Systems (ABS) has been one of the great achievements in automotive vehicle safety. Traditionally, Hydraulically Applied Brakes (HAB) have been the most common system layout in commercial vehicles. Pressure modulation in these systems is generally achieved in a stairway style, making it suitable for threshold-based, fuzzy logic and neural network control [1]. However, alternatives to these systems are now available, such as the Electro-Hydraulic Brake (EHB) or Electro-Mechanical Brake (EMB) systems. These are characterized by a faster response compared with conventional hydraulic systems [2,3] and allow a more precise and continuous control of the braking torque at the wheels.

Many different control strategies have been proposed to address the ABS control. Rule-based algorithms compose the majority of solutions nowadays [4] but, in addition to fuzzy logic [5] and neural network [6,7] controllers, the large amount of tuning parameters make them extremely time-consuming options and are not able to deal with the uncertainties and disturbances of the tire–road dynamics. Moreover, none of these methodologies can assure the stability of the system. Given that brake actuator technology has significantly advanced in the last two decades, researchers have focused their efforts on more advanced control techniques to improve ABS performance. In [8], a robust Integral Sliding Mode Controller (ISMC) was proposed, demonstrating the importance of reference adaptation during braking. Nevertheless, ISMCs are feedback techniques, and adding feed-forward action is not trivial, which limits the performance of the controller. Model Predictive Control has risen as one of the most promising control alternatives [9,10], offering space for improvements with respect to state-of-the-art controllers. However, as MPC algorithms are online strategies for control, the limitation of these systems lies in the computational time required for the correct operation of the algorithm. Sometimes, in fact, the computation time is unpredictable, as the system encounters external disturbances that have not been taken into account in the design, which is a problem for real-time applications where safety is a critical condition. Moreover, the addition of nonconvex constraints also increases the computation time, and the online solvers used in the literature only offer convergence to local optima [11,12]. Classical robust control approaches allow to deal with uncertainties, disturbances and noise by design, while ensuring stability, and do not present the computational drawback of the above, as the control gains for the controller are calculated offline [13,14,15,16,17].

Limited evaluations of robust control techniques are found in the recent literature [1], and existing ones do not validate their results with a high-order vehicle model [18,19] and do not present a simultaneous stage of the vehicle state’s estimation. Motivated by the aforementioned reasons, the design of an $H_{\infty }$ gain-scheduling controller to deal with the tire slip control problem is presented in this paper, and results are validated with the vehicle dynamics software simulator CarSim. The main contributions are:The proposed controller is able to make the tire slip follow a given reference based on the TRFC, resulting in a small reference error and good transient response, guaranteeing system stability. Since the estimation of the TRFC is not the focus of this article, it is assumed to be known for making use of any of the most recent literature algorithms [20,21,22,23,24,25,26,27,28,29,30,31,32].The braking forces are maximized depending on road condition.Even though a simple vehicle model was taken into consideration for the controller design, the proposed algorithm was tested in the vehicle dynamics simulator software CarSim, in which simulations were carried out for different road conditions.To consider the longitudinal velocity and tire–road contact time-dependency problem, a time-varying parameter approach is considered for the synthesis of the controller. These parameters are considered as pseudomeasures.In order to estimate the states of the vehicle and the time-varying parameters with the information obtained from on-board series-production vehicle sensors, a Kalman Filter is considered.

The rest of the article is organized as follows: in Section 2, the problem of the $H_{\infty }$ gain-scheduling controller and vehicle states estimation is depicted. Moreover, the braking problem and dynamics are formulated. In Section 3, the design of the proposed controller is explained. The controller is tested in Section 4 using CarSim and Simulink, and the results obtained are analyzed. Finally, the conclusions are drawn in Section 5.

## 2. Problem Formulation

In this section, the problem of the $H_{\infty }$ gain-scheduling controller and vehicle states estimation is depicted in Figure 1. The vehicle and friction models used for the controller are presented subsequently and all the parameters used are shown in Appendix A.

As shown in Figure 1, a Kalman Filter algorithm is used to estimate the braking tire force of each wheel and the longitudinal velocity of the vehicle. These estimations are then used to calculate the longitudinal slip on each wheel and for the model used by the $H_{\infty }$ controller. To simplify the algorithm, the TRFC is supposed to be obtained by some estimation method [20,21,22,23,24,25,26,27,28,29,30,31,32] and the optimal tire slip that maximizes the braking force is calculated by means of the Burckhardt friction model. Finally, the $H_{\infty }$ controller generates the necessary braking pressure for each wheel in order to minimize the error between the optimal and current longitudinal slip.

### Vehicle and Friction Models

In this section, the vehicle and friction models used for the controller are presented. A single-corner model [33] is used to represent the dynamics of the wheel during braking. It is assumed that the vehicle only moves in the longitudinal direction during the braking maneuver, as in Figure 2.

The dynamics of the single-corner vehicle model depicted in Figure 2 can be expressed as in [33]:(1)$$\left\{\begin{array}{c} J\dot{\omega }=F_{x}R-T_{b}\\ F_{x}=-m{\dot{v}}_{x} \end{array}\right.$$
where *J* is the moment of inertia of the wheel, *m* is the equivalent mass of the single-corner vehicle model and *R* is the effective radius of the wheel; ω is the rotational velocity of the wheel, $T_{b}$ is the braking torque applied on the wheel, $v_{x}$ is the longitudinal velocity of the vehicle and $F_{x}$ is the force originated from the tire–road contact. This force can be determined by means of the expression
(2)$$F_{x}=\mu \left(\lambda \right)F_{z}$$
where $F_{z}$ is the vertical load and μ is the instantaneous tire–road friction coefficient. For a case of straight-line braking, it is considered that μ only depends on the tire slip:(3)$$\lambda =\frac{v_{x}-\omega R}{v_{x}}$$
with λ∈[0,1] and λ=1 meaning that the wheel is locked. In this work, the Burckhardt friction model is used to characterize the tire–road contact behavior. This model allows to obtain the instantaneous friction coefficient for different road condition as a function of the tire slip:(4)$$\mu \left(\lambda \right)=c_{1}\left(1-e^{-c_{2}\lambda }\right)-c_{3}\lambda$$
where the value of the coefficients $c_{1}$, $c_{2}$ and $c_{3}$ only depends on the road condition, resulting in different friction curves [34], as in Figure 3.

By using the Burckhardt friction model, it is simple to know the value of the longitudinal tire slip that maximizes the braking force, shown in Table 1.

By deriving the Equation (Equation 3) and using Equation (Equation 1), the dynamics of the tire slip can be expressed as
(5)$$\dot{\lambda }=\frac{F_{x}}{mv_{x}}\lambda -\left(\frac{1}{m}-\frac{R^{2}}{J}\right)\frac{F_{z}}{v_{x}}+\frac{R}{Jv_{x}}k_{b}P_{b}$$
where $P_{b}$ is the pressure of the hydraulic system, and constant $k_{b}$ comes from $T_{b}=k_{b}P_{b}$.

In Equation (Equation 5), both $F_{x}$ and $v_{x}$ are pseudomeasure time-varying parameters estimated by a Kalman Filter algorithm presented later in the document. To facilitate the design of the controller, the following time-varying parameters are defined:(6)$$\begin{array}{cc} \rho_{1}\left(t\right) & =F_{x},\;\rho_{1}\in \left[\begin{array}{cc} \underline{F_{x}} & \bar{F_{x}} \end{array}\right]\\ \rho_{2}\left(t\right) & =1/v_{x},\;\rho_{2}\in \left[\begin{array}{cc} 1/\bar{v_{x}} & 1/\underline{v_{x}} \end{array}\right] \end{array}$$
where both time-varying parameters $\rho_{1}$ and $\rho_{2}$ are bounded within an upper and a lower bound denoted by “$\bar{*}$” and “$\underline{*}$”, respectively.

By taking x=[λ], $u=[P_{b}]$ and $\rho =\left[\begin{array}{cc} \rho_{1} & \rho_{2} \end{array}\right]$ from Equation (Equation 5), the dynamics of the longitudinal tire slip can be characterized by
(7)$$\dot{x}=A_{0}\left(\rho \right)x+B_{0}\left(\rho \right)u_{c}+D_{0}d$$
where
(8a)$$A_{0}=\frac{\rho_{1}\rho_{2}}{m}$$
(8b)$$B_{0}=\frac{R\rho_{1}k_{b}}{J}$$
(8c)$$D_{0}=1$$
and *d* is considered as the disturbances: $d=-\left(\frac{1}{m}-\frac{R^{2}}{J}\frac{}{}\right)\frac{F_{z}}{v_{x}}$.

## 3. Controller Design

In this section, the proposed $H_{\infty }$ controller synthesis is presented, as well as the proposed algorithm for the vehicle states estimation.

### 3.1. Controller Design Objectives

The main objective of the controller is to make the tire slip ratio follow the desired reference $r=[\lambda_{opt}]$ that maximizes the braking force according to the Burckhardt model, shown in Table 1. Then, the state space of the system expressed in Equation (Equation 7) can be augmented with a new defined state $\zeta =\int_{0}^{t}\left(\lambda -\lambda_{opt}\right)dt$ and $\eta ={\left[\lambda \;\zeta \right]}^{T}$. The dynamics of the augmented system is
(9)$$\dot{\eta }=A\left(\rho \right)\eta +B_{u}\left(\rho \right)u_{c}+B_{d}d+B_{r}r$$
where
(10)$$A\left(\rho \right)=\left[\begin{array}{cc} A_{0} & 0\\ 1 & 0 \end{array}\right],B_{u}\left(\rho \right)=\left[\begin{array}{c} B_{0}\\ 0 \end{array}\right],B_{d}=\left[\begin{array}{c} 1\\ 0 \end{array}\right],B_{r}=\left[\begin{array}{c} 0\\ -1 \end{array}\right]$$

The controlled output of the system is
(11)z=Gη
where $G=\left[\begin{array}{cc} 0 & 1 \end{array}\right]$. The gain controller law proposed for the system in Equation (Equation 9) is of the form
(12)$$u_{c}\left(t\right)=K\left(t\right)\eta$$
and results in a generalized proportional integral controller whose integral term works towards eliminating the error with the reference signal, minimizing the error with respect to the optimal slip ratio.

### 3.2. Stability Analysis

In order to minimize the controlled output, the $H_{\infty }$ performance inequality is chosen as in [35]:(13)$${\left||z|\right|}_{2}^{2}<\gamma_{1}^{2}{\left||r|\right|}_{2}^{2}+\gamma_{1}^{2}\gamma_{2}^{2}{\left||d|\right|}_{2}^{2}$$
and it must be fulfilled for any bounded disturbance *d* and reference signal *r*, where $\gamma_{1}$ is the $H_{\infty }$ performance index and $\gamma_{2}$ is a weighting factor.

**Theorem** **1.**
*For a given state feedback gain K, the closed-loop system defined in (Equation 9) is asymptotically stable and guarantees the $H_{\infty }$ performance described in Equation (Equation 13) if there is a matrix $P=P^{T}≻0$ such that*

(14)
$$\left[\begin{array}{cccc} A_{c}^{T}P+PA_{c} & PB_{r} & PB_{d} & C^{T}\\ {}* & -\gamma_{1}^{2}\gamma_{2}^{2}I & 0 & 0\\ {}* & * & -\gamma_{1}^{2}I & 0\\ {}* & * & * & -I \end{array}\right]≺0$$



**Proof.** By choosing a Lyapunov function of the form
(15)$$V=\eta^{T}P\eta$$
and satisfying V>0 and $\dot{V}<0$ with
(16a)P≻0
(16b)$$A_{c}^{T}P+PA_{c}≺0$$
where $A_{c}$ is the closed-loop system matrix $A_{c}=A+BK$.Now, let us define a cost function as
(17)$$\Delta =\dot{V}+z^{T}z-\gamma_{1}^{2}\gamma_{2}^{2}r^{T}r-\gamma_{1}^{2}d^{T}d$$To guarantee that the inequality of Equation (Equation 14) holds, the cost function defined in Equation (Equation 17) must satisfy
(18)Δ(t)<0,∀t≥0By expressing Δ in matrix form and applying Schur’s complement to Equation (Equation 19), it ensures Equation (Equation 14) to be satisfied, so the proof is concluded.
(19)$$\Delta ={\left(\begin{array}{c} \eta \\ r\\ d \end{array}\right)}^{T}\left(\begin{array}{ccc} A_{c}^{T}P+PA_{c}+C^{T}C & PB_{r} & PB_{d}\\ {}* & -\gamma_{1}^{2}\gamma_{2}^{2}I & 0\\ {}* & * & -\gamma_{1}^{2}I \end{array}\right)\left(\begin{array}{c} \eta \\ r\\ d \end{array}\right)$$□

### 3.3. Gain-Scheduling Feedback Gains Design

As the closed-loop plant of the system is expressed as a function of time-varying parameters ρ in Equation (Equation 9), a polytopic system is generated for describing the dynamics of the system [36]:(20)$$\eta =\sum_{i=1}^{N}\alpha_{i}\left(\rho \right)\left(A_{i}\eta +B_{u,i}+B_{d,i}d+B_{r,i}r\right)$$
where $\alpha_{i}\left(\rho \right)$ are the weighting gains that satisfy $\sum_{i=1}^{N}\alpha_{i}\left(t\right)=1,\alpha \left(t\right)>0$ and N=4 for each of the four vertices that represent the four linear submodels of the generated polytope, as shown in Figure 4. These vertices are built from the upper and lower bounds of the $F_{x}$ and $1/v_{x}$ parameters
(21)$$\left\{\begin{array}{c} \Pi_{1}=\left[A\left(\bar{\rho_{1}},\bar{\rho_{2}}\right),B\left(\bar{\rho_{1}},\bar{\rho_{2}}\right)\right]\\ \Pi_{2}=\left[A\left(\bar{\rho_{1}},\underline{\rho_{2}}\right),B\left(\bar{\rho_{1}},\underline{\rho_{2}}\right)\right]\\ \Pi_{3}=\left[A\left(\underline{\rho_{1}},\bar{\rho_{2}}\right),B\left(\underline{\rho_{1}},\bar{\rho_{2}}\right)\right]\\ \Pi_{4}=\left[A\left(\underline{\rho_{1}},\underline{\rho_{2}}\right),B\left(\underline{\rho_{1}},\underline{\rho_{2}}\right)\right] \end{array}\right.$$

The weighting gains α(t) are calculated using the values of ρ(t) as follows:(22)$$\left\{\begin{array}{c} \alpha_{1}\left(t\right)=\left[|\bar{\rho_{1}}-\rho_{1}\left|\right|\bar{\rho_{2}}-\rho_{2}|/\delta_{\rho }\right]\\ \alpha_{2}\left(t\right)=\left[|\bar{\rho_{1}}-\rho_{1}\left|\right|\rho_{2}-\underline{\rho_{2}}|/\delta_{\rho }\right]\\ \alpha_{3}\left(t\right)=\left[|\rho_{1}-\underline{\rho_{1}}\left|\right|\bar{\rho_{2}}-\rho_{2}|/\delta_{\rho }\right]\\ \alpha_{4}\left(t\right)=\left[|\rho_{1}-\underline{\rho_{1}}\left|\right|\rho_{2}-\underline{\rho_{2}}|/\delta_{\rho }\right] \end{array}\right.$$
where $\delta_{\rho }=\left|\left(\bar{\rho_{1}}-\underline{\rho_{1}}\right)\left(\bar{\rho_{2}}-\underline{\rho_{2}}\right)\right|$.

The values of $\rho_{1}$ and $\rho_{2}$ can be obtained online and, through them, the final feedback controller gain *K* can be obtained as a linear combination of the feedback gain of the $K_{i}$ submodels using
(23)$$K=\sum_{i=1}^{N}\alpha_{i}\left(\rho \right)K_{i}$$

With the polytopic system in Equation (Equation 20) and gain law control in Equation (Equation 12), the controller is asymptotically stable, and the $H_{\infty }$ conditions in Equation (Equation 13) are ensured if there is a definite positive matrix Q, a matrix M and a $\gamma_{1}>0$ that satisfy the LMI
(24)$$\begin{array}{cc} \; & \phi_{i,i}≺0,\;\mathrm{for}\;1\leq i\leq 4\\ \; & \phi_{i,j}+\phi_{j,i}≺0,\;\mathrm{for}\;1\leq i\leq j\leq 4 \end{array}$$
where
(25)$$\left[\begin{array}{cccc} \Lambda_{ij} & B_{r} & B_{d} & QC_{i}^{T}\\ {}* & -\gamma_{1}^{2}\gamma_{2}^{2}I & 0 & 0\\ {}* & * & -\gamma_{1}^{2}I & 0\\ {}* & * & * & -I \end{array}\right]≺0$$
with $\Lambda_{ij}=\left(A_{i}Q+B_{u,i}M_{j}\right)+{\left(A_{i}Q+B_{u,i}M_{j}\right)}^{T}$, and the state feedback gain of each submodel of the corresponding vertex of the polytope is obtained as
(26)$$K_{i}=M_{i}Q^{-1}$$

Proof is shown in [36].

In addition, another constraint is used to limit the maximum control output signal so that the maximum pressure supported by the hydraulic system is not exceeded, thus limiting the braking torque. The limitation of the output signal is performed as in [37], where given positive definite matrices Q and M and a positive scalar ϵ, the maximum control output of the system in Equation (Equation 9) can be limited using the constraint
(27)$$\left[\begin{array}{cc} \frac{1}{\epsilon }X & M\\ {}* & Q \end{array}\right]\geq 0$$
with $X\leq P_{b,max}$.

The objective controller gains are found by solving the minimization problem
(28)$$\begin{array}{cc} \min & \gamma_{1}^{2}\\ \mathrm{subject}\;\mathrm{to} & Q=Q^{T}≻0,X=X^{T},\left(25\right)\;\mathrm{and}\;\left(27\right) \end{array}$$

### 3.4. State Variable Estimation through a Kalman Filter

It is necessary for the control feedback to know the values of the states and the values of ρ to calculate the gains $\alpha_{i}$ of the polytope. Therefore, $F_{x}$, $v_{x}$ and λ have to be estimated. For this purpose, a Kalman Filter is used to estimate the longitudinal velocity and the tire braking forces [38], because it allows to estimate the states of a linear system which cannot be measured directly, in this case tire forces. As the tire forces of every wheel of the vehicle are needed, the estimation is performed using Equation (29) into all the wheels of the vehicle:(29)$$\begin{array}{cc} -m_{t}{\dot{v}}_{x} & =F_{x,fl}+F_{x,fr}+F_{x,rl}+F_{x,rr}\\ J{\dot{\omega }}_{fl} & =F_{x,fl}R-T_{b,fl}\\ J{\dot{\omega }}_{fr} & =F_{x,fr}R-T_{b,fr}\\ J{\dot{\omega }}_{rl} & =F_{x,rf}R-T_{b,rl}\\ J{\dot{\omega }}_{rr} & =F_{x,rr}R-T_{b,rr} \end{array}$$
where $m_{t}$ is the total mass of the vehicle, $T_{b,i}$ is the braking torque and $F_{x,i}$ is the braking tire force of the $i^{th}$ wheel. From Equation (Equation 29), the following state-space model is derived
(30)$$\begin{array}{cc} {\dot{\hat{x}}}_{f} & =A_{f}{\hat{x}}_{f}+B_{f}u_{f}\\ y_{f} & =C_{f}{\hat{x}}_{f} \end{array}$$
where the state variables are $x_{f}={\left[v_{x}\;w_{\mathrm{fl}}\;w_{\mathrm{fr}}\;w_{\mathrm{rl}}\;w_{\mathrm{rr}}\right]}^{T}$, and the control inputs are $u_{f}={\left[T_{b,\mathrm{fl}}\;T_{b,\mathrm{fr}}\;T_{b,\mathrm{rl}}\;T_{b,\mathrm{rr}}\right]}^{T}$, which can be known by means of the controller signals. The measurements are the longitudinal acceleration of the vehicle and the wheel rotation speeds, $y_{f}={\left[w_{\mathrm{fl}}\;w_{\mathrm{fr}}\;w_{\mathrm{rl}}\;w_{\mathrm{rr}}\;a_{x}\right]}^{T}$. All the measurement signals can be obtained using inertia or velocity sensors. Longitudinal acceleration $a_{x}$ can be measured by an Inertial Measurement Unit (IMU) [39], while the angular velocity of each wheel ω can be measured with Wheel Pulse Transducers (WPTs) [40]. Even though longitudinal velocity $v_{x}$ can be measured with an odometer, this can lead to imprecise results; therefore, an estimation of $v_{x}$ seems to be the best choice. By augmenting the system with the tire forces, the new state-space variables vector is ${\bar{x}}_{f}={\left[v_{x}\;w_{\mathrm{fl}}\;w_{\mathrm{fr}}\;w_{\mathrm{rl}}\;w_{\mathrm{rr}}\;F_{x,\mathrm{fl}}\;F_{x,\mathrm{fr}}\;F_{x,\mathrm{rl}}\;F_{x,\mathrm{rr}}\right]}^{T}$, and the state equation of the KF written in discrete form is
(31)$$\begin{array}{cc} {\bar{x}}_{f,k+1} & ={\bar{A}}_{f}{\bar{x}}_{f,k}+{\bar{B}}_{f}u_{f,k}+v_{k}\\ y_{f,k} & ={\bar{C}}_{f}{\bar{x}}_{f,k}+w_{k} \end{array}$$
where
$$\begin{array}{cc} {\bar{A}}_{f} & =\left[\begin{array}{ccccccccc} 0 & 0 & 0 & 0 & 0 & \frac{1}{m_{t}} & \frac{1}{m_{t}} & \frac{1}{m_{t}} & \frac{1}{m_{t}}\\ 0 & 0 & 0 & 0 & 0 & \frac{R}{J} & 0 & 0 & 0\\ 0 & 0 & 0 & 0 & 0 & 0 & \frac{R}{J} & 0 & 0\\ 0 & 0 & 0 & 0 & 0 & 0 & 0 & \frac{R}{J} & 0\\ 0 & 0 & 0 & 0 & 0 & 0 & 0 & 0 & \frac{R}{J}\\ 0 & 0 & 0 & 0 & 0 & 0 & 0 & 0 & 0\\ 0 & 0 & 0 & 0 & 0 & 0 & 0 & 0 & 0\\ 0 & 0 & 0 & 0 & 0 & 0 & 0 & 0 & 0\\ 0 & 0 & 0 & 0 & 0 & 0 & 0 & 0 & 0 \end{array}\right],{\bar{B}}_{f}=\left[\begin{array}{cccc} 0 & 0 & 0 & 0\\ \frac{-1}{J} & 0 & 0 & 0\\ 0 & \frac{-1}{J} & 0 & 0\\ 0 & 0 & \frac{-1}{J} & 0\\ 0 & 0 & 0 & \frac{-1}{J}\\ 0 & 0 & 0 & 0\\ 0 & 0 & 0 & 0\\ 0 & 0 & 0 & 0 \end{array}\right],\\ {\bar{C}}_{f} & =\left[\begin{array}{ccccccccc} 0 & 1 & 0 & 0 & 0 & 0 & 0 & 0 & 0\\ 0 & 0 & 1 & 0 & 0 & 0 & 0 & 0 & 0\\ 0 & 0 & 0 & 1 & 0 & 0 & 0 & 0 & 0\\ 0 & 0 & 0 & 0 & 1 & 0 & 0 & 0 & 0\\ 0 & 0 & 0 & 0 & 0 & \frac{-1}{m_{t}} & \frac{-1}{m_{t}} & \frac{-1}{m_{t}} & \frac{-1}{m_{t}} \end{array}\right],{\bar{D}}_{f}=0_{5\times 4} \end{array}$$
where the time variation is defined using the random walk model, as in [38].

The KF algorithm has two steps: the time update step and measurement update step. In the measurement state step, the algorithm uses the measurement to correct the estimation made in the time update step
(32)$$\begin{array}{cc} {\hat{\bar{x}}}_{f,k} & ={\bar{x}}_{f,k}+K_{k}\left(y_{k}-{\bar{C}}_{f}{\bar{x}}_{f,k}\right)\\ P_{k} & =\left(I-K_{k}{\bar{C}}_{f}\right)P_{k-1}{\left(I-K_{k}{\bar{C}}_{f}\right)}^{T}+K_{k}R_{k}K_{k}^{T} \end{array}$$
where
(33)$$K_{k}=P_{k-1}{\bar{C}}_{f}^{T}\left({\bar{C}}_{f}P_{k-1}{\bar{C}}_{f}^{T}R_{k}\right)$$

In the time update step, an estimation of the state variables is made using the dynamics equations of the system
(34)$$\begin{array}{cc} {\bar{x}}_{f,k+1} & ={\hat{\bar{x}}}_{f,k}\\ P_{k+1} & ={\bar{A}}_{f}P_{k}{\bar{A}}_{f}^{T}+Q_{k} \end{array}$$

The process noise $v_{k}$ is considered to have zero mean and $Q_{k}$ covariance, the measurement noise $v_{k}$ is considered to have zero mean and $R_{k}$ covariance and $P_{k}$ is the states’ covariance. Through these estimations, the tire slip of the wheels can be calculated using Equation (Equation 35). The tire slip is estimated using the measurement of the angular velocity of the wheels and the estimated longitudinal velocity:(35)$${\hat{\lambda }}_{i}=\frac{{\hat{v}}_{x}-\omega_{i}R}{{\hat{v}}_{x}},\;\mathrm{for}\;i=fl,fr,rl,rr$$

## 4. Simulations and Results

### 4.1. Simulation Set Up

This section shows the conditions and results of the simulations performed to test the operation of the $H_{\infty }$ controller designed in the previous section, which is used to control the slip of the four tires of the vehicle. Simulations are carried out in the vehicle dynamics software CarSim, which allows to run simulations with a 27-DOF vehicle model [41]. The controller and state estimator are implemented in Matlab–Simulink. Since during the braking process the vertical load is not the same on both axles of the vehicle due to the load transfer from the rear wheels to the front wheels, one controller is calculated for the rear wheels and another for the front wheels, considering that both the left and right wheels of the same axle work under identical conditions. The gains of the controller are obtained by solving the LMI minimization problem using the *Robust Control Toolbox*.

The limit values for parameters $\rho_{1}$ and $\rho_{2}$ are defined in Table 2. The velocity range considered is 3−19.44m/s. The minimum force on the tire is 0 *N*, and the maximum for the front occurs when the friction coefficient is maximum, considering load transfer. For the case of the rear tire, the maximum forces are calculated when only static load is considered
(36)$$\begin{array}{cc} F_{xmax,front} & =g\mu_{max}\left(m_{f}+\frac{m_{t}h_{cr}\mu_{max}}{2L}\right)\\ F_{xmax,rear} & =g\mu_{max}m_{r} \end{array}$$
where *L* is stated in Table 3. The friction coefficient considered in Equation (Equation 36) is the maximum for the road considered in the simulations, $\mu_{max}=1.00$.

The feedback gains and the $H_{\infty }$ performance index for the front and rear braking controllers are calculated by choosing a weighting factor $\gamma_{2}=1$ in order to take into account the disturbances, shown in Equation (Equation 13). The gain matrices obtained are
(37)$$\begin{array}{cc} K_{1,front} & =\left[-21.6,-1765.2\right],\;K_{1,rear}=\left[-26.6,-1873.2\right]\\ K_{2,front} & =\left[-21.6,-1778.6\right],\;K_{2,rear}=\left[-26.6,-1890.7\right]\\ K_{3,front} & =\left[-32.9,-2699.5\right],\;K_{3,rear}=\left[-40.6,-2874.8\right]\\ K_{4,front} & =\left[-32.9,-2699.9\right],\;K_{4,rear}=\left[-40.6,-2873.9\right]\\ \gamma_{1,front} & =0.0174,\;\gamma_{1,rear}=0.0144 \end{array}$$

The initial, process and measurement covariances for the Kalman Filter are
(38)$$\begin{array}{cc} P_{0} & =Q_{k}=diag{\left\{\begin{array}{ccccccccc} 10^{-7} & 10^{-1} & 10^{-1} & 10^{-1} & 10^{-1} & 5\cdot 10^{2} & 5\cdot 10^{2} & 5\cdot 10^{2} & 5\cdot 10^{2} \end{array}\right\}}^{T}\\ R_{k} & =diag{\left\{\begin{array}{ccccc} 10^{-5} & 10^{-5} & 10^{-5} & 10^{-5} & 10^{-3} \end{array}\right\}}^{T} \end{array}$$
where $R_{k}$ is the covariance considered on the sensors signals.

In order to test the performance of the designed controller, simulations are performed using the vehicular dynamics software CarSim, considering a C-Class vehicle model. This category includes series-production vehicles such as Audi A3, Fiat Bravo or Opel Astra, among others. During the simulation, errors in the sensor measurements are considered. The controller and estimator are implemented the Simulink environment, Figure 1. The controller is tested in different road condition in which the vehicle always starts at a velocity of 70 km/h and starts braking at 0.1 seconds along a straight path. The cut-off speed of the controller is 3 m/s; below this velocity the actuator applies the maximum allowable pressure, as the wheel locking at very low velocities does not compromise the braking maneuver. In all simulations, it is assumed that the friction coefficient $\mu_{max}$ is known, and no error in the estimation is assumed. Hence, the slip reference $\lambda_{opt}$ is obtained by comparing the estimation of $\mu_{max}$ with the closest value from Table 1. The coefficient of friction $\mu_{max}$ is also considered the same for all the wheels; thus, the same reference is always provided to all the controllers. The results are compared with those obtained with a PID controller with gains $K_{P}=10$, $K_{D}=0.5$ and $K_{I}=600$ under the same simulation conditions.

### 4.2. Braking with Constant $\mu_{max}$

The braking maneuver is simulated with the following road conditions:Road condition 1: road with $\mu_{max}=1.00$ trying to emulate a dry asphalt road.Road condition 2: road with $\mu_{max}=0.40$ trying to emulate a wet cobblestone road.Road condition 3: road with $\mu_{max}=0.20$ trying to emulate a snowy road.

The results of this simulations can be seen in Figure 5, Figure 6, Figure 7, Figure 8, Figure 9, Figure 10, Figure 11, Figure 12 and Figure 13. For simplicity, only the results relative to the wheels of the left side of the vehicle are shown.

In Figure 5, Figure 6, Figure 8, Figure 9, Figure 11 and Figure 12, it can be seen that the designed controller manages to make the longitudinal tire slip reach the given reference for the three tested different road conditions better than the PID controller does, especially in the case where the friction coefficient is high, where the proposed controller presents less steady-state error. The settling time is approximately 0.1 seconds in all the simulations, being faster than the PID controller in all the situations.

Figure 7, Figure 10 and Figure 13 show the results of the KF estimations of the tire forces. These estimates are adjusted to the values provided by CarSim.

### 4.3. Braking Test with Changing $\mu_{max}$

In Figure 14, Figure 15 and Figure 16, a snowy stretch on the road where the vehicle brakes is simulated. It can be seen that when the sudden friction change occurs, the controller prevents the slip from increasing too much and thus stopping the wheel from locking. In addition to that, the controller makes the slip of both the front and rear tires follow the reference $\lambda_{opt}$, even though the tires of each axle enter the snowy section at different time instants. The entering and the exit of the car from the snowy patch is pointed out in Figure 14 and Figure 16 with discontinuous lines. Again, the proposed controller performs better than the PID controller, as it has a faster response and minimizes error further.

### 4.4. Braking Distance Comparison

The braking distances obtained using the designed controller are compared with the ones obtained using a PID controller and the default braking ABS that CarSim uses. This system activates and deactivates the brake pressure to maintain the tire slip between two values, 0.1–0.15 for the front wheels and 0.05–0.1 for the rear wheels. The results are shown in Table 4.

## 5. Conclusions and Future Works

In this work, an $H_{\infty }$ gain-scheduling controller able to optimize vehicle braking in an emergency situation was developed, trying to achieve the optimal longitudinal slip value from the Burckhardt tire model that maximizes the braking force for different road conditions. The controller was validated through braking simulations under different road conditions using CarSim and Simulink. It was observed that the controller is able to follow the reference under different road condition and with a reduced response time. In addition, its robustness against the variations that occur in the system during braking was verified, avoiding wheel locks. As part of a future work, communication delays must be taken into account, and an Event-Triggering mechanism should be applied to reduce the network communication loads and actuator chattering, leading to a more complete and realistic braking control.

## Figures and Tables

**Figure 1 sensors-23-01417-f001:**
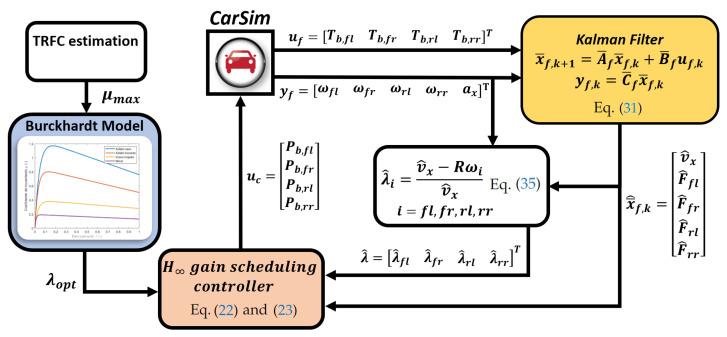
Scheme of the control architecture implemented in Simulink and CarSim [20,21,22,23,24,25,26,27,28,29,30,31,32].

**Figure 2 sensors-23-01417-f002:**
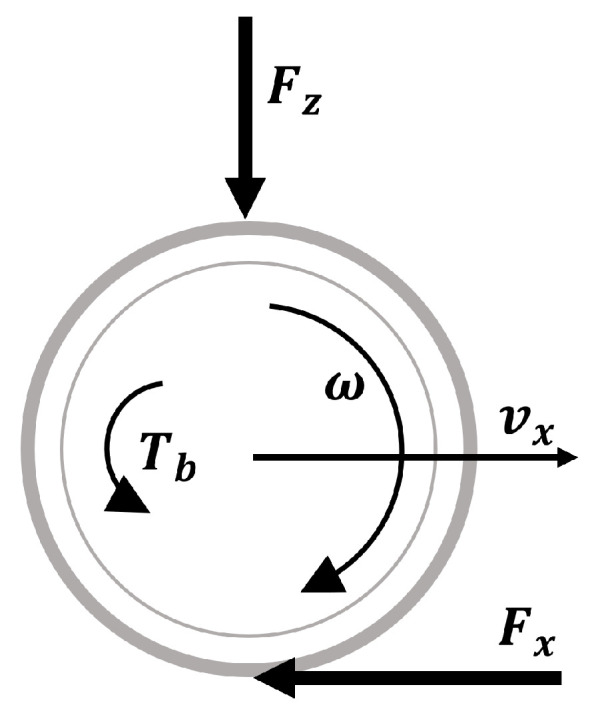
Single-corner vehicle model representation.

**Figure 3 sensors-23-01417-f003:**
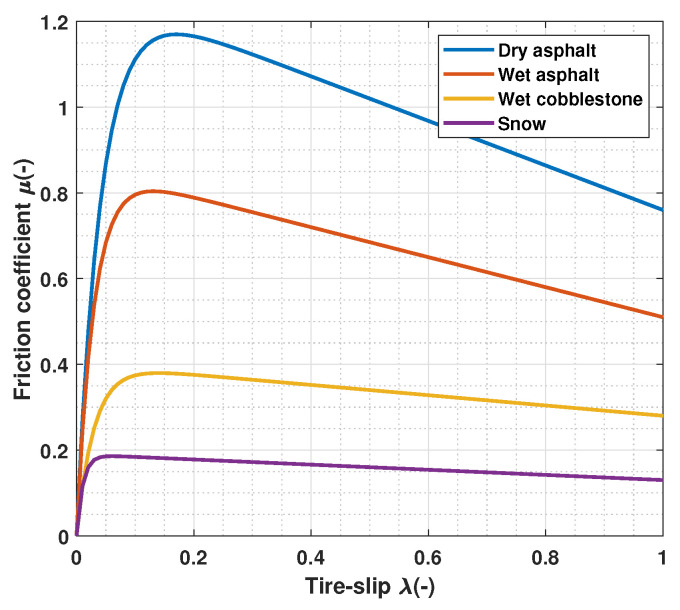
Friction coefficient for different road condition according to the Burckhardt model.

**Figure 4 sensors-23-01417-f004:**
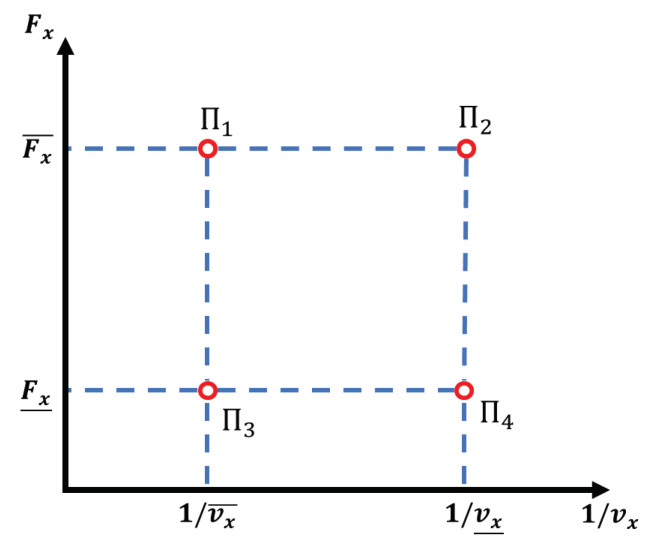
Graphical representation of the four-vertex polytope.

**Figure 5 sensors-23-01417-f005:**
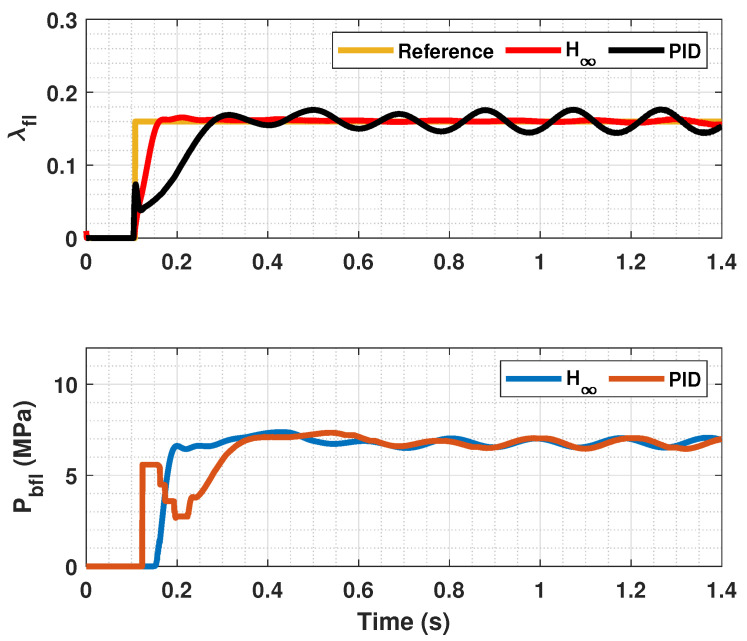
Front tire slip, reference tire slip and brake pressure for $\mu_{max}=1.00$.

**Figure 6 sensors-23-01417-f006:**
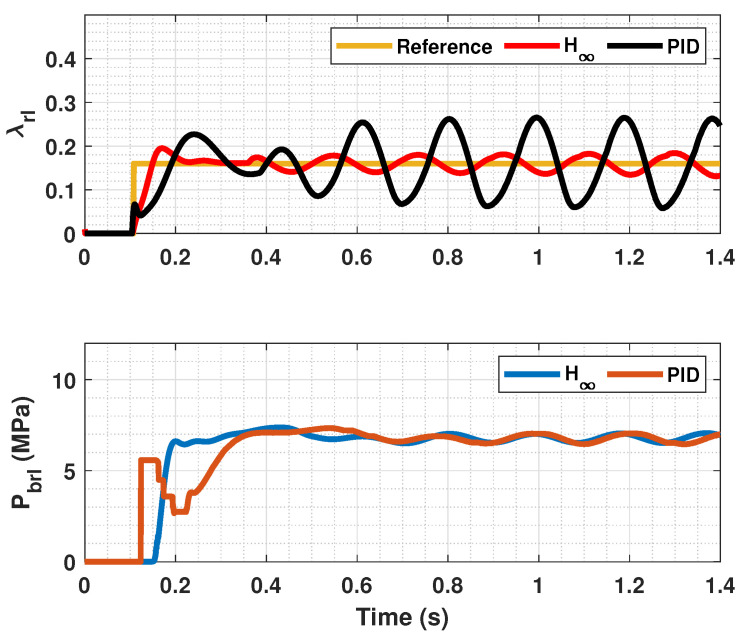
Rear wheels tire slip for front, reference tire slip and brake pressure for $\mu_{max}=1.00$.

**Figure 7 sensors-23-01417-f007:**
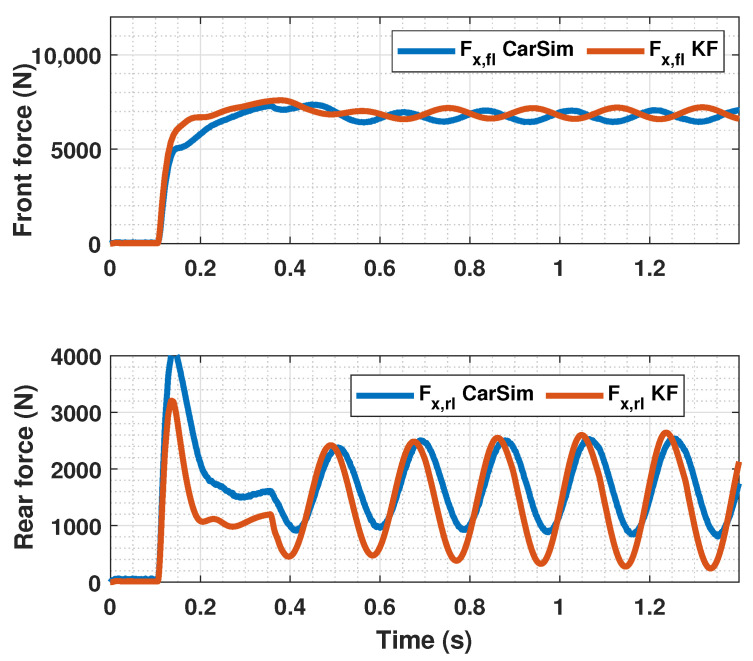
Estimated forces by KF compared with CarSim forces for $\mu_{max}=1.00$.

**Figure 8 sensors-23-01417-f008:**
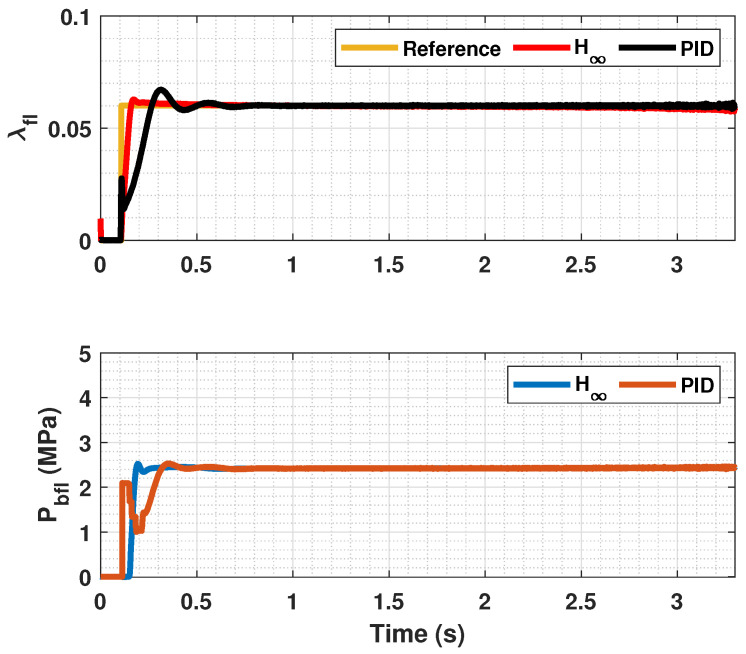
Front wheels tire slip, reference tire slip and brake pressure for $\mu_{max}=0.40$.

**Figure 9 sensors-23-01417-f009:**
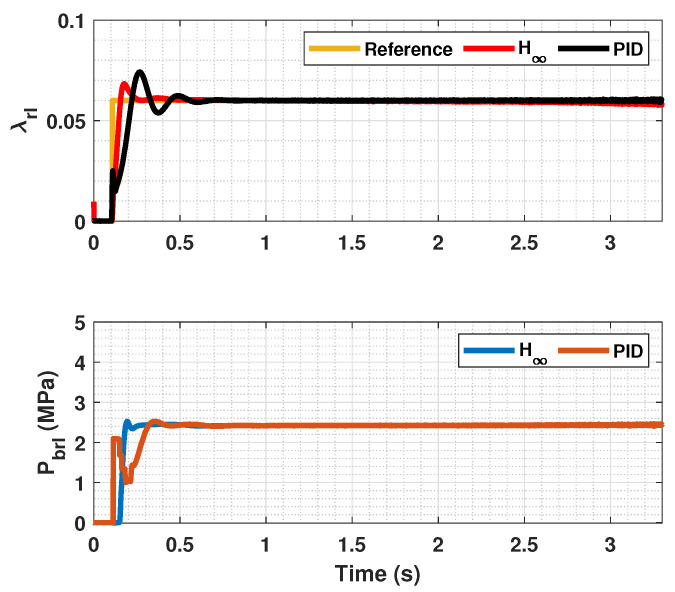
Rear wheels tire slip, reference tire slip and brake pressure for $\mu_{max}=0.40$.

**Figure 10 sensors-23-01417-f010:**
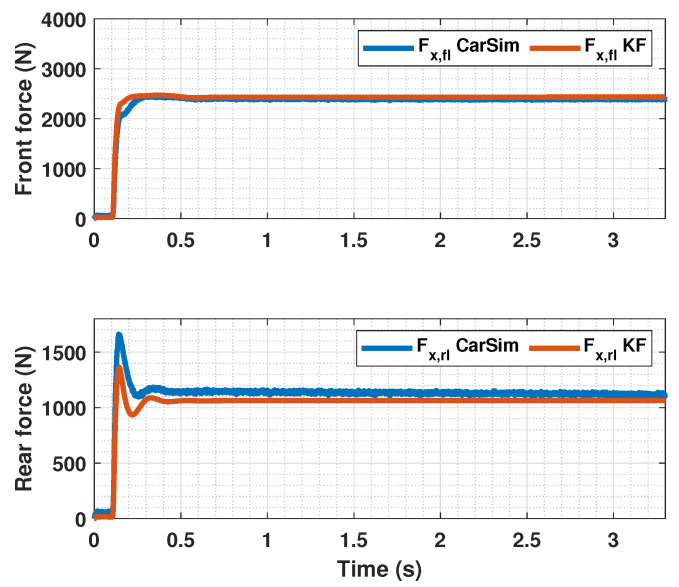
Estimated forces by KF compared with CarSim forces for $\mu_{max}=0.40$.

**Figure 11 sensors-23-01417-f011:**
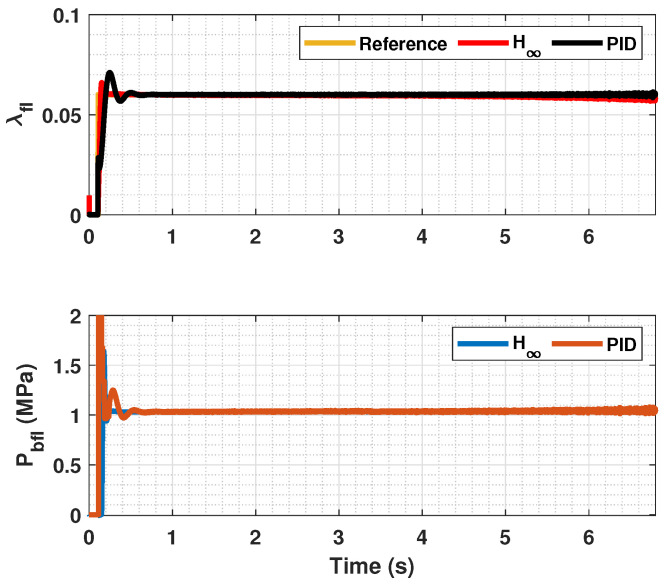
Front wheels tire slip, reference tire slip and brake pressure for $\mu_{max}=0.20$.

**Figure 12 sensors-23-01417-f012:**
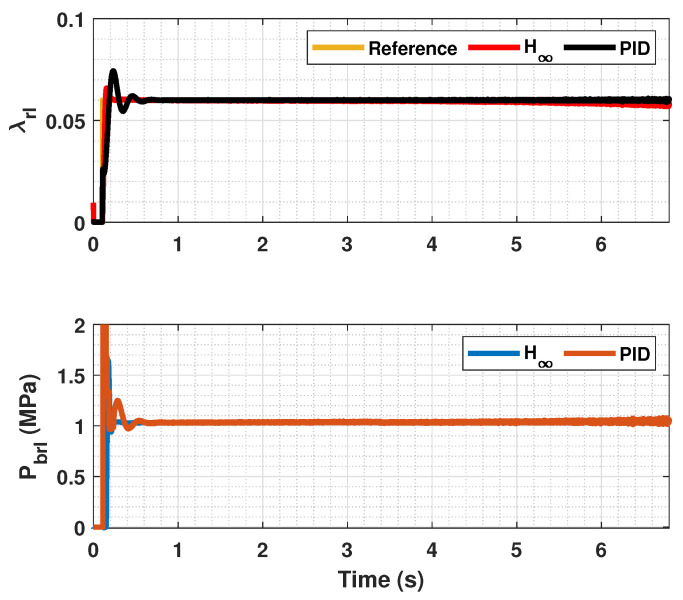
Rear wheels tire slip, reference tire slip and brake pressure for $\mu_{max}=0.20$.

**Figure 13 sensors-23-01417-f013:**
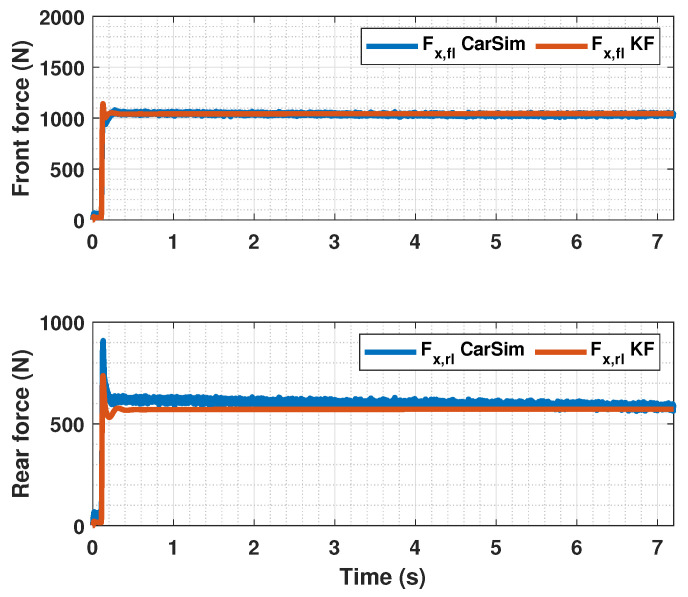
Estimated forces by KF compared with CarSim forces for $\mu_{max}=0.20$.

**Figure 14 sensors-23-01417-f014:**
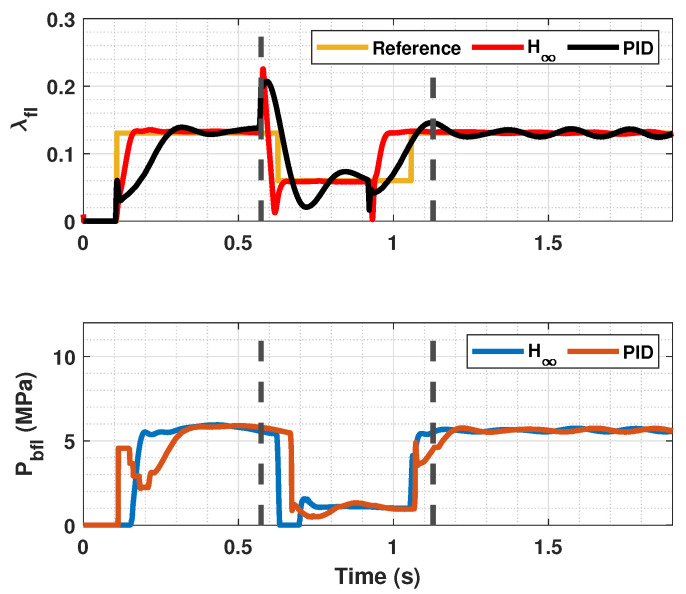
Front wheels tire slip, reference tire slip and brake pressure for changing $\mu_{max}$.

**Figure 15 sensors-23-01417-f015:**
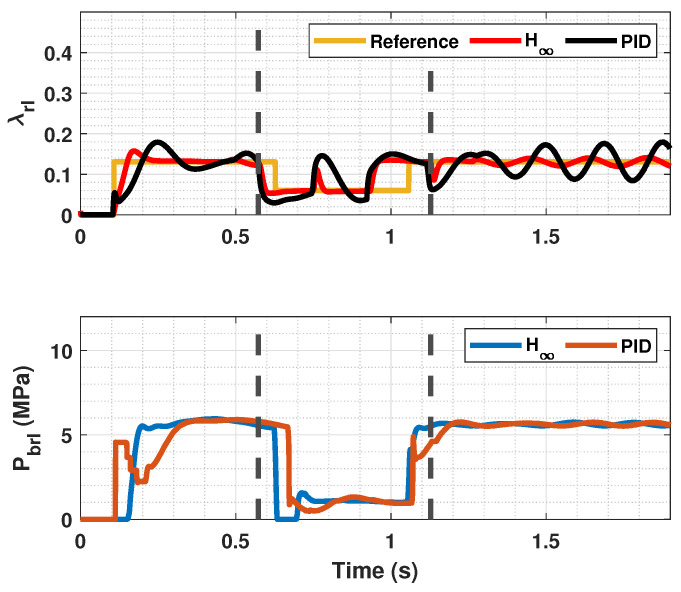
Rear wheels tire slip, reference tire slip and brake pressure for changing $\mu_{max}$.

**Figure 16 sensors-23-01417-f016:**
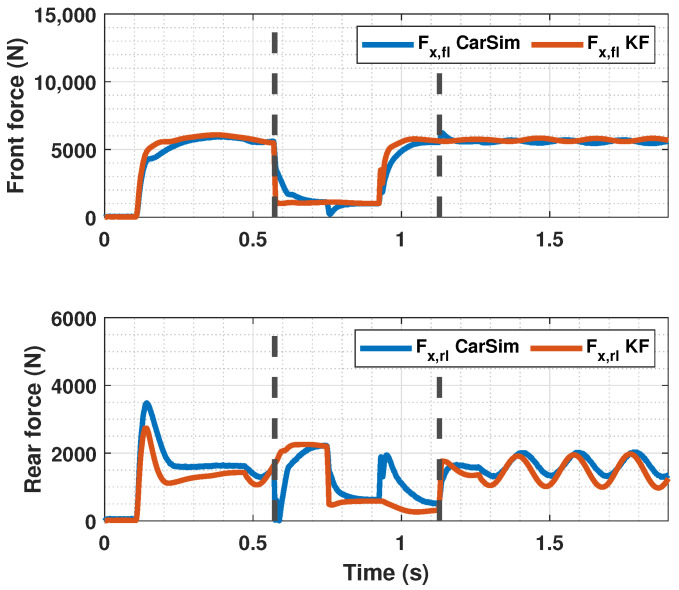
Estimated forces by KF compared with CarSim forces for changing $\mu_{max}$.

**Table 1 sensors-23-01417-t001:** Burckhardt friction model parameters [34].

Burckhardt Parameters Values					
**Road Condition**	$c_{1}$	$c_{2}$	$c_{3}$	$\mu_{\max}$	$\lambda_{opt}$
**Dry asphalt**	1.280	23.990	0.520	1.170	0.170
**Wet asphalt**	0.857	33.820	0.350	0.800	0.130
**Wet cobblestone**	0.400	33.710	0.120	0.380	0.140
**Snow**	0.195	94.130	0.060	0.190	0.060

**Table 2 sensors-23-01417-t002:** Polytopes bounds.

Polytope Bounds		
**Parameter**	**Front Controller**	**Rear Controller**
$\bar{\rho_{1}}$	5601 N	2737 N
$\underline{\rho_{1}}$	0 N	0 N
$\bar{\rho_{2}}$	0.33 s/m	0.33 s/m
$\underline{\rho_{2}}$	0.0514 s/m	0.0514 s/m

**Table 3 sensors-23-01417-t003:** Vehicle characteristics.

Vehicle Characteristics			
**Parameter**	**Definition**	**Value**	**Units**
$m_{f}$	Front wheel equivalent mass	428.97	kg
$m_{r}$	Rear wheel equivalent mass	279.03	kg
$m_{t}$	Vehicle mass	1416	kg
$k_{bf}$	Front braking constant	300	Nm/MPa
$k_{br}$	Rear braking constant	200	Nm/MPa
$P_{b,max}$	Maximum brake pressure	10	MPa
*J*	Spin inertia	0.9	kgm${}^{2}$
*R*	Wheel radius	0.31	m
*L*	Vehicle wheelbase	2.578	m
$h_{cr}$	Center of gravity height	0.35	m

**Table 4 sensors-23-01417-t004:** Braking distances comparison.

Braking Distance (m)			
**Road: $\mu_{\max}$**	**$H_{\infty }$ Controller**	**PID**	* **CarSim** * **ABS**
1.00	16.21	16.47	17.35
0.40	38.24	38.79	46.38
0.20	81.55	82.65	93.91
0.85→0.20→0.85	22.68	23.20	24.04

## Data Availability

Not applicable.

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
