# Peer review of "Tire Slip H∞ Control for Optimal Braking Depending on Road Condition"

_sensors, 2023, doi:10.3390/s23031417_

Round 1
Reviewer 1 Report
This work presents a new controller in order to control the tire slip in vehicle braking on different road conditions. To the understanding of this reviewer, this work is within the scope of this journal and significantly interesting. The quality of the presented work is high, the methods are presented clearly and the results support the conclusions extracted.
This manuscript should be subject of minor revisions regarding the written english (e.g. only (line 99) appears two times, constrain should be constraint when it is a noun)
Reviewer 2 Report
In this paper, the authors investigated a new approach to deal with the tire slip control problem and maximize the braking forces depending on the road condition. The paper is well written, and its organization is appropriate; however, the derivation and explanation of some steps are sometimes very brief and therefore difficult to follow. Here are the reviewer’s detailed comments:
1. I think it is better to include all abbreviations and variables in a table.
2. A reference should be added to model 1 and table 1.
3. All variables in model 29 should be identified, and a reference should be added.
4. The state variable estimation using an EKF needs more details. Why is such an estimator used? How to design the parameters of the filter?
5. The limitation of the proposed approach should be added.
6. It is better to compare the proposed control scheme with existing ones.
7. future work should be mentioned in the conclusion.
8. Literature review should be enhanced, considering the following references
https://doi.org/10.1016/j.ejcon.2017.10.005
https://doi.org/10.1080/00207721.2016.1266527
DOI: 10.1109/ITSC.2013.6728347
Reviewer 3 Report
This paper studied a model-based control for an anti-lock braking system (ABS). The CarSim simulation results are used to demonstrate the enhancement of the proposed controller in decreasing the braking distance. However, some major issues need to be addressed before publication.
1) Different types of ABS controls and different groups of estimations on state estimation methods are categorized in the Introduction. However, some details are missing. The reviewer has some questions as follows:
1-1) In paragraph 3; it is mentioned that “The computational load of MPCs hampers the applicability of these controllers in current on-board CPUs. Moreover, none of these methodologies can assure the stability of the system” Now I have two concerns about this sentence: The linear models as used in (9) face a closed-form solution for MPC which is easy to implement. On the other hand, as I see, the stability of the MPC-based controllers is stated clearly in most earlier research.
1-2) In paragraph 4; different estimation methods are discussed. However, in the motivation section it is stated that “Since the estimation of the TRFC is not the focus of this article, it is assumed to be known from the use of some of the references discussed above [22]-[27]”. What is the purpose of the author for reviewing the estimation methods?
2) What is the research motivation? The motivation section should be thoroughly improved.
3) After Eq. (6), it is stated that the time-varying parameters are bounded. During the braking procedure, what is the bound of \rho_2 when v_x converges to zero?
4) For estimation of the longitudinal force of the tires, a model such as random walk is defined for the force of the tires. Therefore, the state of the force model is augmented by the states of the system. How did you estimate the force of the tire? Since you mentioned them as your states. The estimation method section should be thoroughly improved.
5) The CarSim simulator is used to verify the proposed control method. Now I have some questions:
5-1) Why didn’t you use the half-car model for your modeling? How did you arrange the load transfer issue?
5-2) The selected model of the car should be mentioned in the simulation section.
5-3) How did you tune the ABS control settings in the CarSim model? Are the default parameters selected?
5-4) The simplifications and assumptions should be completely mentioned.
